# Efficient Synthesis of New Fluorinated β-Amino Acid Enantiomers through Lipase-Catalyzed Hydrolysis

**DOI:** 10.3390/molecules25245990

**Published:** 2020-12-17

**Authors:** Sayeh Shahmohammadi, Ferenc Fülöp, Enikő Forró

**Affiliations:** 1Institute of Pharmaceutical Chemistry, University of Szeged, H-6720 Szeged, Hungary; sayeh.s@pharm.u-szeged.hu (S.S.); fulop@pharm.u-szeged.hu (F.F.); 2Stereochemistry Research Group of the Hungarian Academy of Sciences, University of Szeged, H-6720 Szeged, Hungary

**Keywords:** kinetic resolution, lipase-catalyzed hydrolysis, enantioselective synthesis, fluorinated β-amino acid

## Abstract

An efficient and novel enzymatic method has been developed for the synthesis of β-fluorophenyl-substituted β-amino acid enantiomers through lipase PSIM (*Burkholderia cepasia*) catalyzed hydrolysis of racemic β-amino carboxylic ester hydrochloride salts **3a**–**e** in *i*Pr_2_O at 45 °C in the presence of Et_3_N and H_2_O. Adequate analytical methods were developed for the enantio-separation of racemic β-amino carboxylic ester hydrochlorides **3a**–**e** and β-amino acids **2a**–**e**. Preparative-scale resolutions furnished unreacted amino esters (*R*)-**4a**–**e** and product amino acids (*S*)-**5a**–**e** with excellent *ee* values (≥99%) and good chemical yields (>48%).

## 1. Introduction

In recent years, enantiomerically pure β-aryl-substituted β-amino acids have been intensively investigated due to their pharmacological significance, unique and remarkable biological properties [1], their utility in synthetic chemistry [2], and drug research [3]. Therefore, this class of compounds has been documented as a crucial scaffold in the design and synthesis of conceivable pharmaceutical drugs. For instance, 3-amino-3-phenylpropionic acid, which is a key pharmaceutical building block, is present in anticancer agents, such as Taxol [4]. It can also find application as a fundamental component in the synthesis of novel antibiotics [5] and analgesic endomorphine-1 analogue tetrapeptides [6].

On the other hand, tremendous achievements in the development of fluorinated amino acid drugs verified the high importance of this type of compounds in pharmaceutical chemistry. It is known that the occurrence of fluorine in biologically active natural compounds is extremely low. In turn, the number of fluorine-containing drugs on the market is rising continuously. The reasons are the unique characteristics of the fluorine atom in terms of its high electronegativity and the polarity of a carbon–fluorine bond [7,8]. Thus, incorporation of fluorine into β-amino acids has gained increasing attention in recent decades. For example, Januvia (sitagliptin phosphate) acts as an antidiabetic agent via inhibition of dipeptidyl peptidase IV [9], whereas (±)-Eflornithine was used for the treatment of trypanosomiasis [10] and against facial hirsutism in women [11].

There are different approaches for the synthesis of optically active β-aryl-β-amino acids [12,13,14]. The utilization of enzymes in these reactions gained special attention, which is due to their ability to conduct the reactions enantio-selectively. For example, lipases are stable. They work under mild conditions and many of them are commercially available. They can be applied on an industrial scale [15,16]. Lipase-catalyzed methods for the resolution of both cyclic [17] and acyclic [18] β-amino carboxylic esters through hydrolysis are known in the literature. Various enzymatic procedures have been developed by our research group for the preparation of biologically active β-aryl-substituted, β-heteroaryl-substituted, and β-arylalkyl-substituted β-amino acid enantiomers through enantioselective (*E* > 200) hydrolysis of the corresponding β-amino carboxylic esters both in H_2_O or in an organic solvent catalyzed by lipase (*Pseudomonas cepacia*) PS[19,20,21]. Catalyzed kinetic and dynamic kinetic resolution of β-amino carboxylic esters or their hydrochloride salts with tetra-hydro-isoquinoline and tetra-hydro-β-carboline skeleton through hydrolysis have been performed. Catalysts used include *Candida antarctica* lipase B (CAL-B) (in aqueous NH_4_OAc buffer at pH 8.5 and or in *i*Pr_2_O in the presence of 1 equiv of H_2_O), *Alcalase* (in borate buffer at pH 8), and lipase PS (in *i*Pr_2_O with 4 equiv of added H_2_O) [22,23,24,25].

A number of N-benzylated β^2^-, β^3^-, and β^2,3^-amino acids were prepared through CAL-B-catalyzed hydrolysis of the corresponding racemic amino carboxylic esters with H_2_O in n-hexane or 2-methyl-2-butanol, under stirring [26] or utilizing high-speed ball-milling conditions [27]. Covalently immobilized lipase AK (*Pseudomonas fluorescens*) and lipase PS were used as efficient stereoselective catalysts for the kinetic resolution of exotic and variously substituted *rac*-(5-phenylfuran-2-yl)-β-alanine ethyl ester hydrochlorides through hydrolysis (*E* > 146) in NH_4_OAc buffer (20 mM, pH 5.8) at 30 °C [28]. Gotor et al. reported the kinetic resolution (*E* > 200) of a large number of 3-amino-3-phenylpropanoate esters through lipase PS-catalyzed hydrolysis with H_2_O in 1,4-dioxane at 45 °C [18]. The method was successfully used for the synthesis of (*S*)-3-amino-3-phenylpropionic acid, which is a key precursor for the preparation of (*S*)-dapoxetine, and a potent selective serotonin reuptake inhibitor (SSRI) used for the treatment of depression, bulimia, or anxiety [29]. In addition, very recently, Zhang et al. summarized a review of the most facile catalytic enantioselective strategies to construct the fluorinated α-amino and β-amino acids [30].

Herein, in view of the importance of fluorinated β-amino acids, our aim was to synthesize (±)-β-amino carboxylic ester hydrochloride salts **3a**–**e** (Scheme 1), then to devise a suitable enzymatic protocol for the synthesis of new fluorinated β-amino acids via enantioselective hydrolysis of **3a**–**e** (Scheme 2) and provide an adequate characterization of the enantiomeric products.

## 2. Results

### 2.1. Synthesis of Ethyl 3-Amino-3-Arylpropanoate Hydrochloride Salts (±)-***3a**–**e***

Racemic β-amino acids (±)-**2a**–**e** were synthesized by modified Rodionov synthesis through the reaction of the corresponding aldehydes with malonic acid in the presence of NH_4_OAc in EtOH at a reflux temperature (Scheme 1) [31,32]. Subsequently, the β-amino carboxylic ester hydrochloride salts (±)-**3a**–**e** were prepared with yields ranging from 76% to 98% by esterification of the corresponding β-amino acids in the presence of SOCl_2_ in EtOH.

### 2.2. Enzyme-Catalyzed Hydrolysis of (±)-***3a**–**e***

#### 2.2.1. Preliminary Experiments

On the basis of the results achieved on the enzyme-mediated enantioselective hydrolysis of β-amino carboxylic esters [17,18], the hydrolysis of model compound (±)-**3a** (Scheme 2) was conducted with 5 equiv of Et_3_N and 0.5 equiv of H_2_O in the presence of 30 mg mL^–1^ enzyme in *i*Pr_2_O at 45 °C (Table 1, entry 1). In the frame of enzyme screening, lipase AY (*Candida rugosa*), lipase AK, PPL (*Porcine pancreatic* lipase), and CAL-B (Table 1, entries 2–5) showed activity in enzymatic hydrolysis. However, with the exception of lipase AK affording an *ee*_p_ value of 75% and a moderate *E* (8) (19% conversion in 10 min, entry 3), low reactivities and low enantio-selectivities were achieved (entries 2, 4, and 5). It is noteworthy that PPL catalyzed the reaction with opposite enantio-selectivity. Lipase PSIM, in contrast, provided an *E* value of 108 (entry 1) and, consequently, it was selected for further studies.

Next, we analyzed the effect of solvent on enantio-selectivity and reaction rate. Very different *E* and reaction rate data were observed in the green solvents tested (Table 2). The hydrolytic reactions of **3a** in the ether-type solvents were rapid (conv. 52%, 51%, and 54% after 10 min, *E* = 59, 113, and 27, entries 1, 2, and 4), while, in EtOAc, the reaction proceeded relatively slowly with low enantioselectivity (conv. 11%, after 10 min, *E* = 3, entry 3). On the basis of our earlier results [33], the reaction was also performed under solvent-free conditions when, in harmony with our earlier observation, a reasonable enantioselectivity (*E* 74) was observed in addition to a rapid transformation (conv. 49% after 10 min, entry 5). For the reason of economy (taking into account that 2-Me-THF is the most expensive selected solvent), despite the highest *E* (113), *i*Pr_2_O, with a slightly lower *E* (108), but as significantly less expensive was identified as the most suitable solvent.

In order to follow up the progress of the reaction while maintaining high enantioselectivity, it was wise to slow down the reaction. When the reaction temperature decreased from 45 °C to 25 °C, both the reaction rate and enantio-selectivity for the hydrolysis of **3a** clearly decreased (30% conv. in 10 min, *E* = 48 vs. 48% conv. in 10 min, *E* = 108, Table 1, entry 1). To our surprise, the fastest reaction was achieved at 3 °C with the highest degree of conversion (50% in 10 min) and an *E* value of 134. In order to collect more information, we decided to carry out the reaction with different enzyme concentrations at 3 °C. As shown in Table 3, there was no significant difference in the reaction rates, when the enzyme concentration decreased from 10 to 5 or 2 mg mL^–1^ (~50% conv. in 10 min reaction time, entries 1–3). In contrast, *E* dropped significantly when the reaction was performed with a 2-mg mL^–1^ enzyme (entry 3). Since both high *E* and satisfactory reaction rate were attained at 45 °C, we decided to use this optimal reaction temperature for preparative-scale reactions. Additionally, a set of preliminary experiments was performed in order to determine the influence of enzyme concentration on the reaction rate (Table 4). The reaction rate for the hydrolysis of (±)-**3a** clearly increased as the concentration of enzymes was increased. The highest reaction rate was observed with a 40-mg mL^–1^ enzyme (entry 5). However, for a satisfactory reaction time (the time needed to reach 50% conversion), the use of a 30-mg mL^–1^ enzyme (Table 1, entry 1) was selected for preparative-scale resolutions.

#### 2.2.2. Preparative-Scale Resolutions of (±)-3**a**–**e**

Preparative-scale resolution of (±)-3**a**–**e** were performed under the optimized conditions (30 mg mL^–1^ lipase PSIM, 0.5 equiv Et_3_N, 0.5 equiv H_2_O, *i*Pr_2_O, 45 °C) to yield the unreacted β-amino carboxylic ester enantiomers (*R*)-4a–e and product β-amino acids (*S*)-5**a**–**e** with excellent *ee* (≥99%) and good yields (>48%) (Table 5).

#### 2.2.3. Determination of Absolute Configurations

The absolute configurations in the cases of (*S*)-**5a** {[α] = −3.1 (*c* 0.28, H_2_O), lit. [36] [α] = −3.9 (*c* 1.0, H_2_O)}, (*S*)-**5b** {[α] = −5.0 (*c* 0.26, H_2_O), lit. [36] [α] = −4.5 (*c* 0.50, H_2_O)}, (*S*)-**5c** {[α] = −3.0 (*c* 0.28, H_2_O), lit. [37] [α] = +3.9 (*c* 0.40, H_2_O for the antipode (*R*))}, (*R*)-**4a** {[α] = +17.9 (*c* 0.44, CHCl_3_), lit. [36] [α] = −21.5 (*c* 1.0, CHCl_3_ for antipode (*S*))}, (*R*)-**4b** {[α] = +9.0 (*c* 0.29, CHCl_3_), lit. [36] [α] = −12.6 (*c* 0.50, CHCl_3_ for antipode (*S*))} and (*R*)-**4c** {[α] = +18.9 (*c* 0.41, CHCl_3_), lit. [18] [α] = +18.5 (*c* 1.0, CHCl_3_)} were assigned by comparing the [α] values with literature data. Taking into consideration the most stable conformation of the 3-amino-3-phenylpropanoate core matching nicely, the (*S*)-configuration of the products [18] and the GC chromatograms analyzed, the same enantio-preference for the (*S*)-selective hydrolysis by Lipase PSIM for **5d** and **5e** was indicated.

## 3. Experimental Section

### 3.1. General Methods

Lipase PSIM and lipase AK were from Amano Pharmaceuticals and lipase AY was from Fluka. PPL and CAL-B immobilized on acrylic resin were purchased from Sigma (Budapest, Hungary). Substituted benzaldehydes were from Sigma-Aldrich. Triethylamine was from Merck. Solvents of the highest analytical grade were from Sigma-Aldrich. Optical rotations were measured with a Perkin-Elmer 341 Polarimeter. ^1^H-NMR spectra were recorded on a Bruker Avance DRX 500 spectrometer. Melting points were determined on a Kofler apparatus (see the Appendix A). The enantiomeric excess *ee* values for the unreacted β-amino carboxylic ester and the β-amino acid enantiomers produced were determined by GC equipped with a Chirasil-L-Val column after double derivatization [34] with (i) diazomethane [Caution: the derivatization with diazomethane should be performed under a well-working hood] and (ii) acetic anhydride in the presence of 4-dimethylaminopyridine and pyridine [90 °C for 10 min → 170 °C (temperature rise 20 °C min^−1^), 10 psi]. Retention times (min) for **4a**: 36.308 (antipode: 36.535), for **5a**: 32.365 (antipode: 31.515), for **4b**: 32.137 (antipode: 32.550), for **5b**: 29.031 (antipode: 28.282), for **4c**: 33.305 (antipode: 33.860), for **5c**: 29.905 (antipode: 29.528), for **4d**: 26.064 (antipode: 26.187), for **5d**: 23.766 (antipode: 23.463), for **4e**: 35.421 (antipode: 36.018), for **5e**: 30.946 (antipode: 30.541)].

### 3.2. General Procedure for the Syntheses of Racemic β-Amino Acids ***2a**–**e***

Compounds **2a**–**e** were synthesized based on the modified Rodionov synthesis [31,32] through condensation of the corresponding aldehydes **1a**–**e** (2 g) with malonic acid (1 equiv) in the presence of NH_4_OAc (2 equiv) in EtOH under reflux for 8 h [19]. The resulting white crystals were filtered off and washed with acetone and then they were recrystallized from H_2_O and acetone.

#### 3.2.1. (±)-3-Amino-3-(3,4-Difluorophenyl) Propionic Acid **2a**

Yield: (1.2 g, 42%), mp 233–235 °C. ^1^H-NMR (D_2_O, 500 MHz): δ = 7.30–7.26 (m, 1H, Ar), 7.25–7.21 (m, 1H, Ar), 7.15–7.13 (m, 1H, Ar), 4.54 (t, *J* = 7.33 Hz, 1H, CH), 2.78 (dd, *J*_AB_ = 16.31 Hz, *J*_AX_ = 7.89 Hz, 1H, C(2)H_A_), 2.70 (dd, *J*_BA_ = 16.27 Hz, *J*_BX_ = 6.78 Hz, 1H, C(2)H_B_).

#### 3.2.2. (±)-3-Amino-3-(3,5-Difluorophenyl) Propionic Acid **2b**

Yield: (1.36 g, 48%), mp 237–239 °C. ^1^H-NMR (D_2_O, 500 MHz): δ = 6.98–6.97 (m, 2H, Ar), 6.93–6.89 (m, 1H, Ar), 4.56 (t, *J* = 7.23 Hz, 1H, CH), 2.78 (dd, *J*_AB_ = 16.34 Hz, *J*_AX_ = 7.84 Hz, 1H, C(2)H_A_), 2.71 (dd, *J*_BA_ = 16.34 Hz, *J*_BX_ = 6.61 Hz, 1H, C(2)H_B_).

#### 3.2.3. (±)-3-Amino-3-(4-Fluorophenyl) Propionic Acid **2c**

Yield: (1.50 g, 51%), mp 244–246 °C. ^1^H-NMR (D_2_O, 500 MHz): δ = 7.37–7.34 (m, 2H, Ar), 7.11–7.07 (m, 2H, Ar), 4.54 (t, *J* = 7.31 Hz, 1H, CH), 2.81 (dd, *J*_AB_ = 16.10 Hz, *J*_AX_ = 8.05 Hz, 1H, C(2)H_A_), 2.72 (dd, *J*_BA_ = 16.10 Hz, *J*_BX_ = 6.73 Hz, 1H, C(2)H_B_).

#### 3.2.4. (±)-3-Amino-3-(2-Fluoro-4-Triflouromethylphenyl) Propionic Acid **2d**

Yield: (0.786 g, 30%), mp 251–253 °C. ^1^H-NMR (D_2_O, 500 MHz): δ = 7.75–7.72 (m, 1H, Ar), 7.62–7.57 (m, 2H, Ar), 4.89 (t, *J* = 7 Hz, 1H, CH), 2.82 (dd, *J*_AB_ = 16.63 Hz, *J*_AX_ = 8.63 Hz, 1H, C(2)H_A_), 2.77 (dd, *J*_BA_ = 16.86 Hz, *J*_BX_ = 5.31 Hz, 1H, C(2)H_B_).

#### 3.2.5. (±)-3-Amino-3-(2-Fluoro-4-Methylphenyl) Propionic Acid **2e**

Yield: (0.4275 g, 15%), mp 239–241 °C. ^1^H-NMR (D_2_O, 500 MHz): δ = 7.39–7.36 (m, 1H, Ar), 7.11–7.03 (m, 2H, Ar), 4.84 (t, *J* = 7.78 Hz, 1H, CH), 2.80 (dd, *J*_AB_ = 16.81 Hz, *J*_AX_ = 9.82 Hz, 1H, C(2)H_A_), 2.69 (dd, *J*_BA_ = 16.75 Hz, *J*_BX_ = 4.24 Hz, 1H, C(2)H_B_), 2.38 (s, 3H, CH_3_).

### 3.3. General Procedure for the Syntheses of Racemic β-Amino Carboxylic Ester Hydrochloride Salts ***3a**–**e***

SOCl_2_ (1.05 equiv) was added to 30 mL of EtOH at a temperature kept under –15 °C with saline ice. To this solution, **2a**–**e** (1 g) were added at once. The mixture was stirred at 0 °C for 30 min, then at room temperature for 3 h, and, finally, heated under reflux for 1 h. The solvent was evaporated off under reduced pressure and the resulting white **3a**–**e**. HCl salts were recrystallized from EtOH and Et_2_O.

#### 3.3.1. Hydrochloride Salt of Ethyl (±)-3-Amino-3-(3,4-Difluorophenyl) Propanoate **3a**. HCl

Yield: (1 g, 76%), mp 142–144 °C. ^1^H-NMR (D_2_O, 500 MHz): δ = 7.32–7.28 (m, 1H, Ar), 7.27–7.23 (m, 1H, Ar), 7.18–7.16 (m, 1H, Ar), 4.69 (t, *J* = 8.73 Hz, 1H, CH), 4.05–4.01 (m, 2H, CH_2_), 3.08 (dd, *J*_AB_ = 16.77 Hz, *J*_AX_ = 7.20 Hz, 1H, C(2)H_A_), 3.00 (dd, *J*_BA_ = 16.77 Hz, *J*_BX_ = 7.50 Hz, 1H, C(2)H_B_), 1.05 (t, *J* = 7.09 Hz, 3H, CH_3_). ^13^C-NMR (D_2_O, 126 MHz): δ = 13.2, 37.9, 50.7, 62.5, 116.6 (d, ^2^*J*_C–F_ = 18.42 Hz), 118.4 (d, ^2^*J*_C–F_ = 17.65 Hz), 124.1 (dd, ^3^*J*_C–F_ = 7.10 Hz, ^4^*J*_C–F_ = 3.64 Hz), 132.2 (d, ^3^*J*_C–F_ = 3.87 Hz), 150.2 (dd, ^1^*J*_C–F_ = 250.46 Hz, ^2^*J*_C–F_ = 16.46 Hz), 150.6 (dd, ^1^*J*_C–F_ = 241.81 Hz, ^2^*J*_C–F_ = 5.74 Hz), 171.3. ^19^F-NMR (D_2_O, 471 MHz): δ = −136.6 Hz, −136.9 Hz.

#### 3.3.2. Hydrochloride Salt of Ethyl (±)-3-Amino-3-(3,5-Difluorophenyl) Propanoate **3b**. HCl

Yield: (1.15 g, 87%), mp 182–184 °C. ^1^H-NMR (D_2_O, 500 MHz): δ = 7.02–7.01 (m, 2H, Ar), 6.97–6.93 (m, 1H, Ar), 4.74 (t, *J* = 7.20 Hz, 1H, CH), 4.07–4.02 (m, 2H, CH_2_), 3.11 (dd, *J*_AB_ = 16.98 Hz, *J*_AX_ = 7.24 Hz, 1H, C(2)H_A_), 2.81 (dd, *J*_BA_ = 16.96 Hz, *J*_BX_ = 7.23 Hz, 1H, C(2)H_B_), 1.08 (t, *J* = 6.98 Hz, 3H, CH_3_). ^13^C-NMR (D_2_O, 126 MHz): δ = 15.6, 40.2, 53.2, 65.0, 107.6 (t, ^2^*J*_C–F_ = 25.43 Hz), 113.0 (dd, ^2^*J*_C–F_ = 20.28 Hz, ^4^*J*_C–F_ = 6.70 Hz), 141.0 (t, ^3^*J*_C–F_ = 9.47 Hz), 165.6 (dd, ^1^*J*_C–F_ = 248.20 Hz, ^3^*J*_C–F_ = 13.04 Hz), 173.7. ^19^F-NMR (D_2_O, 471 MHz): δ = −108.3 Hz.

#### 3.3.3. Hydrochloride Salt of Ethyl (±)-3-Amino-3-(4-Flourophenyl) Propanoate **3c**. HCl

Yield: (1.11 g, 82%), mp 165–167 °C. ^1^H-NMR (D_2_O, 500 MHz): δ = 7.39–7.37 (m, 2H, Ar), 7.13–7.09 (m, 2H, Ar), 4.70 (t, *J* = 7.82 Hz, 1H, CH), 4.04–3.99 (m, 2H, CH_2_), 3.09 (dd, *J*_AB_ = 16.61 Hz, *J*_AX_ = 7.17 Hz, 1H, C(2)H_A_), 3.00 (dd, *J*_BA_ = 16.66 Hz, *J*_BX_ = 7.65 Hz, 1H, C(2)H_B_), 1.05 (t, *J* = 7.21 Hz, 3H, CH_3_). ^13^C-NMR (D_2_O, 126 MHz): δ = 13.3, 38.3, 51.1, 62.6, 116.3 (d, ^2^*J*_C–F_ = 22.08 Hz), 129.5 (d, ^3^*J*_C–F_ = 8.94 Hz), 131.3 (d, ^4^*J*_C–F_ = 2.99 Hz), 163.2 (d, ^1^*J*_C–F_ = 246.30 Hz),171.6. ^19^F-NMR (D_2_O, 471 MHz): δ = −112.3 Hz.

#### 3.3.4. Hydrochloride Salt of Ethyl (±)-3-Amino-3-(2-Fluoro-4-Trifluoromethylphenyl) Propanoate **3d**. HCl

Yield: (1.2 g, 95%), mp 133–135 °C. ^1^H-NMR (D_2_O, 500 MHz): δ = 7.61–7.58 (m, 1H, Ar), 7.56–7.53 (m, 2H, Ar), 5.03 (t, *J* = 7.31 Hz, 1H, CH), 4.07–4.02 (m, 2H, CH_2_), 3.20 (dd, *J*_AB_ = 16.90 Hz, *J*_AX_ = 7.27 Hz, 1H, C(2)H_A_), 3.12 (dd, *J*_BA_ = 16.94 Hz, *J*_BX_ = 7.35 Hz, 1H, C(2)H_B_), 1.07 (t, *J* = 7.51 Hz, 3H,CH_3_). ^13^C-NMR (D_2_O, 126 MHz): δ = 13.1, 36.9, 45.8 (d, ^3^*J*_C–F_ = 2.69 Hz), 62.6, 113.9 (dq, ^2^*J*_C–F_ = 25.50 Hz, ^3^*J*_C–F_ = 3.80 Hz), 123 (q, ^1^*J*_C–F_ = 274.12 Hz), 122.2 (m), 126.3 (d, ^2^*J*_C–F_ = 12.87 Hz), 129.8 (d, ^3^*J*_C–F_ = 3.44 Hz), 133.2 (m), 160.0 (d, ^1^*J*_C–F_ = 248.95 Hz), 171.1. ^19^F-NMR (D_2_O, 471 MHz): δ = −62.6 Hz, −114.9 Hz.

#### 3.3.5. Hydrochloride Salt of Ethyl (±)-3-Amino-3-(2-Fluoro-4-Methylphenyl) Propanoate **3e**. HCl

Yield: (1.30 g, 98%), mp 172–174 °C. ^1^H-NMR (D_2_O, 500 MHz): δ = 7.26–7.23 (m, 1H, Ar), 7.03–6.99 (m, 2H, Ar), 4.90 (t, *J* = 7.51 Hz, 1H, CH), 4.05–4.01 (m, 2H, CH_2_), 3.15 (dd, *J*_AB_ = 16.55 Hz, *J*_AX_ = 7.32 Hz, 1H, C(2)H_A_), 3.06 (dd, *J*_BA_ = 16.57 Hz, *J*_BX_ = 7.73 Hz, 1H, C(2)H_B_), 2.25 (s, 3H, CH_3_), 1.06 (t, *J* = 7.12 Hz, 3H, CH_3_,). ^13^C-NMR (D_2_O, 126 MHz): δ = 13.1, 20.3, 37.2, 46.1 (d, ^3^*J*_C–F_ = 2.77 Hz), 62.5, 116.6 (d, ^2^*J*_C–F_ = 21.22 Hz), 118.8 (d, ^2^*J*_C–F_ = 13.22 Hz,),125.8 (d, ^4^*J*_C–F_ = 2.85 Hz), 128.3 (d, ^3^*J*_C–F_ = 3.71 Hz), 143.3 (d, ^3^*J*_C–F_ = 8.42 Hz), 160.1 (d, ^1^*J*_C–F_ = 245.57 Hz), 171.5. ^19^F-NMR (D_2_O, 471 MHz): δ = −118.5 Hz.

### 3.4. General Procedure for the Preparative-Scale Resolutions of (±) ***3a**–**e***

Racemic hydrochloride salts **3a**–**e** (200 mg) were dissolved in *i*Pr_2_O (10 mL). Lipase PSIM (30 mg mL^−1^), Et_3_N (5 equiv), and H_2_O (0.5 equiv) were added and the mixture was shaken in an incubator shaker at 45 °C for Rt: 8 h **3a**, 72 h **3b**, 18 h **3c**, 26 h **3d**, 23 h **3e** (Table 5). Reactions were stopped by filtering off the enzyme at close to 50% conversion. The filtered enzyme was washed with Et_2_O (3 × 15 mL). The solvents were dried by using Na_2_SO_4_, and then evaporated off to yield unreacted β-amino carboxylic esters (*R*)-**4a**–**e**. The filtered enzyme was washed with distilled H_2_O (3 × 15 mL). Then evaporation of the filtrate yielded the crystalline (*S*)-**5a**–**e** products, which where recrystallized from EtOH and H_2_O. All of the enantiomers formed in enzymatic reactions were isolated in basic form due to a relatively slow in situ liberation of basic amino ester from its hydrochloric salt, which is followed by enzymatic hydrolysis.

#### 3.4.1. (*R*)-Ethyl 3-Amino-3-(3,4-Difluorophenyl) Propanoate **4a**

Yield: (84 mg, 48.7%), [α] = +17.9 (*c* 0.44, CHCl_3_), lit. [36] [α] = –21.5 (*c* 1.0, CHCl_3_ for antipode (*S*)), [α] = +4.1 (*c* 0.33, EtOH). The ^1^H-NMR (D_2_O, 500 MHz) spectroscopic data were similar to those for **3a**.

#### 3.4.2. (*R*)-Ethyl 3-Amino-3-(3,5-Difluorophenyl) Propanoate **4b**

Yield: (66 mg, 38.26%), [α] = +9.0 (*c* 0.29, CHCl_3_), lit. [36] [α] = –12.6 (*c* 0.50, CHCl_3_ for antipode (*S*)), [α] = –5.0 (*c* 0.20, EtOH). The ^1^H-NMR (D_2_O, 500 MHz) spectroscopic data were similar to those for **3b**.

#### 3.4.3. (*R*)-Ethyl 3-Amino-3-(4-Fluorophenyl) Propanoate **4c**

Yield: (83 mg, 49%), [α] = +18.9 (*c* 0.41, CHCl_3_), lit. [18] [α] = +18.5 (*c* 1.0, CHCl_3_), [α] = +12.8 (*c* 0.32, EtOH). The ^1^H-NMR (D_2_O, 500 MHz) spectroscopic data were similar to those for **3c**.

#### 3.4.4. (*R*)-Ethyl 3-Amino-3-(2-Fluoro-4-Triflouromethylphenyl) Propanoate **4d**

Yield: (84.6 mg, 47.83%), [α] = +20.3 (*c* 0.53, CHCl_3_), [α] = +13.7 (*c* 0.30, EtOH). The ^1^H-NMR (D_2_O, 500 MHz) spectroscopic data were similar to those for **3d**.

#### 3.4.5. (*R*)-Ethyl 3-Amino-3-(2-Fluoro-4-Methylphenyl) Propanoate **4e**

Yield: (94.8 mg, 47.4%), [α] = +16.0 (*c* 0.13, CHCl_3_), [α] = +6.0 (*c* 0.21, EtOH). The ^1^H-NMR (D_2_O, 500 MHz) spectroscopic data were similar to those for **3e**.

#### 3.4.6. (*S*)-3-Amino-3-(3,4-Difluorophenyl) Propionic Acid **5a**

Yield: (72.8 mg, 48.08%), [α] = −3.1 (*c* 0.28, H_2_O), lit. [36] [α] = −3.9 (*c* 1.0, H_2_O), mp 229–231 °C, lit. [38] mp 226–230 °C. The ^1^H-NMR (D_2_O, 500 MHz) spectroscopic data were similar to those for **2a**.

#### 3.4.7. (*S*)-3-Amino-3-(3,5-Difluorophenyl) Propionic Acid **5b**

Yield: (73 mg, 48.22%), [α] = −5.0 (*c* 0.26, H_2_O), lit. [36] [α] = −4.5 (*c* 0.50, H_2_O), mp 256–258 °C. The ^1^H-NMR (D_2_O, 500 MHz) spectroscopic data were similar to those for **2b**.

#### 3.4.8. (*S*)-3-Amino-3-(4-Fluorophenyl) Propionic Acid **5c**

Yield: (71.7 mg, 48.5%), [α] = −3.0 (*c* 0.28, H_2_O), lit. [37] [α] = +3.9 (*c* 0.40, H_2_O for antipode (*R*)), mp 242–244 °C, lit. [37] mp 245–247 °C. The ^1^H-NMR (D_2_O, 500 MHz) spectroscopic data were similar to those for **2c**.

#### 3.4.9. (*S*)-3-Amino-3-(2-Fluoro-4-Trifluoromethylphenyl) Propionic Acid **5d**

Yield: (77.5 mg, 48.7%), [α] = −11.0 (*c* 0.19, MeOH), mp 255–257 °C. The ^1^H-NMR (D_2_O, 500 MHz) spectroscopic data were similar to those for **2d**.

#### 3.4.10. (*S*)-3-Amino-3-(2-Fluoro-4-Methylphenyl) Propionic Acid **5e**

Yield: (84.8 mg, 48.42%), [α] = −13.0 (*c* 0.21, MeOH), mp 258–260 °C. The ^1^H-NMR (D_2_O, 500 MHz) spectroscopic data were similar to those for **2e**.

## 4. Conclusions

Novel fluorine-containing amino acid enantiomers have been prepared through the hydrolysis of racemic β-amino carboxylic ester hydrochloride salts **3a**–**e** catalyzed by lipase PSIM. Excellent enantioselectivities (*E* > 200) were obtained when the reactions were performed with H_2_O as a nucleophile in *i*Pr_2_O at 45 °C, in the presence of Et_3_N. Both unreacted amino carboxylic esters (*R*)-**4a**–**e** and product amino acids (*S*)-**5a**–**e** were isolated with excellent *ee* (usually ≥99%) and good yields (>48%). Suitable analytical methods were devised to follow the enzymatic reactions and calculate the enantiomeric excess, conversions, and enantio-selectivities.

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
