# Peer review of "Efficient Synthesis of New Fluorinated β-Amino Acid Enantiomers through Lipase-Catalyzed Hydrolysis"

_molecules, 2020, doi:10.3390/molecules25245990_

Round 1

Reviewer 1 Report

Enico Forro and co-workers reports the synthesis of enantiomerically enriched fluorinated β-amino acids through lipase-catalyzed hydrolysis. The synthesis of amino acids via this method is well known in literature and the authors have broad experience in the preparation of chiral compounds using enzymes.

Additional question:

  • why obtained compounds were characterized only by 1H NMR spectra? Since you get fluorinated compounds, I would expect 19F NMR spectra and 13C spectra showing fluorine-carbon couplings.

The following correction should be done by the authors before publication:

  • Scheme 1 should be corrected: (+-) symbol must be removed from 1a-e (aldehydes are achiral); first step of the synthesis – malonic acid and NH4OAc should be added above arrow, EtOH, reflux should go below arrow; protons on phenyl rings are not necessary, they should be removed.
  • Table 4: numbers of substates (3a to 3e) should be bold.
  • You write, for example, optical rotation = -5 (line 141), I think it should be = -5.0, similarly = -3 (same line), line 143, 231, 232, 241, 247, 250, 254, 257,.
  • In all spectra J values should be written immediately after the multiplicity of the signal.
  • In lines: 238, 241 the second optical rotation is probably literature value, please mark this.

Reviewer 2 Report

Dear authors,

Shahmohammadi et al. describe in this article an access to enantiopure β-fluorophenyl-substituted β-amino acid through an enzymatic hydrolysis of racemic β-amino carboxylic ester hydrochloride salts. After a screening of enzymes, solvents and enzyme concentrations on a model substrate 3a, the authors have selected experimental conditions and applied them to a preparative-scale resolution of the model substrate 3a and four other substrates 3b-e. In all cases, the authors obtained good yields and excellent enantiomeric excesses for the unreacted β-amino carboxylic esters enantiomers (R) and the product β-amino acids enantiomers (S).

Even if this method is efficient and good results have been obtained for the five substrates studied in this article, this manuscript is a simple extension of related studies (on β-aryl-β-aminoacids: ref. 19-21, 23 and on tetrahydroisoquinoline aminoacids: ref. 22, 24-25) with a lack of novelty and originality on a very limited scope (only five substrates have been studied). Moreover, the choice of the optimized conditions doesn’t appear to be clear from the preliminary investigations: i) Me-THF seems to be better than iPr2O, but the authors have selected the iPr2O (table 2, lines 104-105). ii) The fastest reaction was obtained at 3 °C (unfortunately not included in the table 2) but they decided to perform the reaction at 45 °C. iii) finally, they showed that the amount of enzyme can be reduced to 5mg.mL-1, but they illogically performed the preparative-scale reactions with 30 mg.mL-1.

For all these reasons, I am not approving this manuscript for publication in the journal Molecules. More examples and more clear explanations on the choice of selected conditions need to be added to make this manuscript suitable for publication

Additional remarks:

In the scheme 1, the conditions of the first transformation are not properly indicated on the scheme (malonic acid and ammonium acetate are missing).

Lines 342, reference 30: “1-32” are not the final page numbers.

Reviewer 3 Report

A succinct article describing a useful synthetic methodology. Please consider the following corrections.

1) Add a reference to your calculation of conversion from enantiomeric excess (i.e. postnote 'd' in Tables 1-3). For example https://doi.org/10.1002/bit.260450611

2) Prior to Table 4, e.g. paragraph starting line 130, explain the reaction durations used in the prep scale experiments and why they are different. Do they correspond to the conclusion of the reaction? Also in this paragraph, it is claimed the conditions are optimised, despite superior conditions (temperature, solvent) previously identified but not used. Can you add something to clarify why the ("sub-optimal") prep scale conditions were used, e.g. easier set up, more stable, more reproducible, etc.

Round 2

Reviewer 2 Report

Dear authors,

This revised version provides some of the additional required data but, there is still missing information in the manuscript regarding the choice of the optimized conditions. It is important to give an answer to the referee in the cover letter, but the information should be communicated to any reader of the article. While the selection of iPr2O has been clearly exposed (although not highlighted), the result obtained with 30 mg mL–1 of enzyme at 3 °C, Conv. 50%, E = 134 (line 110) is still not include in any table and there is no sentence explaining why the preparative reactions were done at 45 °C. Furthermore, if the authors had chosen to perform the reaction at 45 °C, why they decided to carry out the reactions with different enzyme concentrations at 3°C, these investigations should have been done at 45 °C.

For all these reasons, I am not approving this manuscript in present form. Additional explanations and experiments on the choice of selected conditions need to be added to make this manuscript suitable for publication.
